# Costs and cost-effectiveness of cervical cancer screening strategies in women living with HIV in Burkina Faso: The HPV in Africa Research Partnership (HARP) study

Angela Devine[1,2]*, Alice Vahanian[3], Bernard Sawadogo[4], Souleymane Zan[5], Fadima Yaya Bocoum[6], Helen Kelly[3], Clare Gilham[3], Nicolas Nagot[7], Jason J. Ong[3,8], Rosa Legood[3], Nicolas Meda[4], Alec Miners[3], Philippe Mayaud[3], on behalf of the HARP Consortium[¶]

1 Global and Tropical Health Division, Menzies School of Health Research, Charles Darwin University, Darwin, Northern Territory, Australia, 2 Centre for Epidemiology and Biostatistics, Melbourne School of Population and Global Health, University of Melbourne, Victoria, Australia, 3 London School of Hygiene & Tropical Medicine, London, United Kingdom, 4 Centre de Recherche Internationale pour la Santé, Université de Ouagadougou, Ouagadougou, Burkina Faso, 5 Centre Hospitalier Universitaire Yalgado Ouédraogo, Ouagadougou, Burkina Faso, 6 Institut de Recherche en Sciences de la Santé, Ouagadougou, Burkina Faso, 7 Pathogenesis and control of chronic infections, INSERM, Etablissement Francais du Sang, University of Montpellier, Montpellier, France, 8 Central Clinical School, Monash University, Melbourne, Victoria, Australia

¶ Membership of the HARP Consortium is provided in the Acknowledgments
* angela.devine@menzies.edu.au

**Data Availability Statement:** All relevant data are within the paper.

## Abstract

### Introduction

This study estimated the costs and incremental cost per case detected of screening strategies for high-grade cervical intraepithelial neoplasia (CIN2+) in women living with HIV (WLHIV) attending HIV clinics in Burkina Faso.

### Methods

The direct healthcare provider costs of screening tests (visual inspection with acetic acid (VIA), VIA combined visual inspection with Lugol's iodine (VIA/VILI), cytology and a rapid HPV DNA test (*care*HPV)) and confirmatory tests (colposcopy, directed biopsy and systematic four-quadrant (4Q) biopsy) were collected alongside the HPV in Africa Research Partnership (HARP) study. A model was developed for a hypothetical cohort of 1000 WLHIV using data on CIN2+ prevalence and the sensitivity of the screening tests. Costs are reported in USD (2019).

### Results

The study enrolled 554 WLHIV with median age 36 years (inter-quartile range, 31–41) and CIN2+ prevalence of 5.8%. The average cost per screening test ranged from US$3.2 for VIA to US$24.8 for cytology. Compared to VIA alone, the incremental cost per CIN2+ case detected was US$48 for VIA/VILI and US$814 for *care*HPV. Despite higher costs, *care*HPV

**Funding:** The research leading to these results has received funding from the European Commission (EC) 7th Framework Programme under grant agreement No. HEALTH-2010-F2-265396 (https://ec.europa.eu/research/fp7) awarded to PM. The funder did not contribute in the study design, data collection and analysis, decision to publish, or preparation of the manuscript.

**Competing interests:** The authors have declared that no competing interests exist. The careHPV and Digene HC-II kits used in this project were obtained through the QIAGEN Corporation donation program to the London School of Hygiene & Tropical Medicine. This does not alter our adherence to PLOS ONE policies on sharing data and materials.

was more sensitive for CIN2+ cases detected compared to VIA/VILI (97% and 56%, respectively). The cost of colposcopy was US$6.6 per person while directed biopsy was US$33.0 and 4Q biopsy was US$48.0.

## Conclusion

Depending on the willingness to pay for the detection of a case of cervical cancer, decision makers in Burkina Faso can consider a variety of cervical cancer screening strategies for WLHIV. While *care*HPV is more costly, it has the potential to be cost-effective depending on the willingness to pay threshold. Future research should explore the lifetime costs and benefits of cervical cancer screening to enable comparisons with interventions for other diseases.

## Introduction

In Burkina Faso, like in many countries in sub-Saharan Africa (SSA), invasive cervical cancer is the leading cause of female cancer mortality [1]. The current cervical cancer screening modality in Burkina Faso is visual inspection using acetic acid with the possibility to add Lugol's iodine (VIA/VILI), but information on uptake is sparse and perceived to be low [2], and effectiveness to prevent cancer remains unmeasured in the Burkinabe context. Compared to HIV-negative women, women living with HIV (WLHIV) are at higher risk of persistent high-risk human papillomavirus (HPV) infections, which are aetiologically linked to the development of precursor cervical intraepithelial neoplasia (CIN) lesions, ultimately leading to increased risk of invasive cervical cancer (ICC) [3,4]. HIV care programmes in SSA have seldom been designed to integrate cervical cancer screening, despite ICC being considered an AIDS-defining illness [5]. Thanks to increasing access to antiretroviral therapy (ART), WLHIV have increased life expectancy; and recent studies suggest that early ART, with sufficient duration of use accompanied with sustained HIV viral suppression, may reduce incidence of ICC by as much as 60% [6].

While cytology based on stained cervical scrapes or smears (Papanicolaou method) has been the traditional method for screening in high income countries, HPV DNA testing has been increasingly advocated because of its high sensitivity to detect high-grade cervical intraepithelial neoplasia (CIN2+) [7]. These methods complemented by high-level clinical services offering colposcopy, biopsy for confirmatory histological diagnosis and management of cervical lesions once detected have been credited with considerable declines in cervical cancer mortality in high-income countries [8] where rates are seven to 10 times lower than low-income countries [1]. Cervical cancer screening programmes using cytology or HPV DNA tests are more difficult to implement and sustain in low and middle income countries (LMICs) due to logistical, financial and human resource constraints [9], although efforts are being made in some LMICs to introduce HPV based testing [10]. At the time of this study, Burkina Faso, had only three hospitals (one public) in Ouagadougou, the capital city, that offered the Papanicolaou test [11]. Cervical cancer screening was opportunistic and population-based programmes were not available.

In order to expand access to cervical cancer screening, less resource-intensive techniques are being employed, such as VIA/VILI that can be performed by trained nurses and midwives who can immediately treat lesions with cryotherapy (screen and treat approach). Another

option is triage using lab-based HPV DNA tests such as *care*HPV (QIAGEN Inc, Gaithesburg, MD) [12–16], which involves a second screening test because of its low specificity. Visual inspection has shown to have reasonable sensitivity and specificity for CIN2+/CIN3+ in the general population [17], but can be highly variable depending on setting and operator training and experience [18]. It is unclear if diagnostic accuracy of visual inspection is dependent on HIV status, as few studies evaluating diagnostic accuracy by HIV status report conflicting findings [19,20]. The pooled sensitivity of VIA for CIN2+ in HIV-negative women in SSA has been shown to vary from 76% to 87% [17]. The *care*HPV DNA test has had satisfactorily high and equivalent sensitivity in various settings among HIV-seronegative women [21] and in WLHIV [22] but lower specificity to distinguish CIN2+ among WLHIV due to higher prevalence of HR-HPV in WLHIV [22,23].

To date there has been limited costing evidence on cervical screening in SSA [9,24,25], and the cost-effectiveness of the *care*HPV test against histological outcomes has only been assessed in a handful of countries [14,16,26]. A recent systematic review of the cost-effectiveness of cervical cancer screening strategies in LMICs (not limited to WLHIV) indicated that visual inspection and HPV testing are more cost-effective than cytology and that the cost-effectiveness of HPV testing was dependent on the test costs and performance of visual inspection [9]. The frequent and intensive follow-up of patients at HIV clinics may provide a valuable opportunity to organise cervical cancer screening. Targeting WLHIV is a commonly recommended policy option in SSA [27]; however, this has been limited in implementation. Few studies have investigated the costs and cost-effectiveness of integrating cervical cancer screening into HIV services, and none of these studies have been conducted in Burkina Faso [24,26]. For example, a cost study in Kenya indicated that integrating cervical screening through VIA, VIA/VILI, careHPV, Papanicolaou, or Hybrid Capture II (HC-II) into HIV services would be less costly than a non-integrated programme due to economies of scope (i.e. efficiencies due to broadening services offered) [24].

Our study reports the costs of screening (VIA, VIA/VILI, cytology, HPV) and confirmatory tests (colposcopy, histology) from a health provider perspective in Burkina Faso. The incremental cost per additional CIN2+ case detected using screening or triage strategies for WLHIV attending HIV clinics in Burkina Faso is also evaluated.

## Methods

A cross-sectional study of cervical cancer screening strategies was conducted among WLHIV aged between 25 and 50 years who were enrolled from the Hopital de Jour HIV clinic of the Centre Hospitalier Universitaire Yalgado Ouédraogo (CHU-YO) between November 2011 and April 2012. CHU-YO is the largest HIV clinic in the country, located in Ouagadougou, Burkina Faso. The HPV in Africa Research Partnership (HARP) methodology has been described elsewhere [28]. The HARP study was given approval by the research ethics committees of the Wits University in South Africa (no. 110707), the Ministry of Health in Burkina Faso (no. 2012-12-089), and the London School of Hygiene & Tropical Medicine (no. 7400). Written informed consent was obtained at the screening visit when eligibility for the study was assessed and at enrolment. In brief, all eligible and consenting women were screened using VIA, VILI, *care*HPV, and cytology in one visit. Eligibility criteria were women aged 25–50 years who were HIV-1 seropositive, resident in the city, who had not had any treatment for cervical cancer or hysterectomy, who were not pregnant or less than 8 weeks postpartum. For cytology, 'test-positivity' was considered if atypical squamous cells of undetermined significance, or greater (ASCUS+). Thresholds for test-positivity using low-grade squamous intraepithelial lesions (LSIL) and high-grade squamous intraepithelial lesions (HSIL) were also

considered. All participants, regardless of screening test results, were invited for colposcopy and women showing any anomaly on any of the screening tests or during colposcopic examination were subjected to systematic four-quadrant (4Q) cervical biopsy including directed biopsy of any suspect area to obtain the diagnosis of CIN by histology. Women found to have no anomaly for all tests did not undergo biopsy and were considered CIN negative. In this paper, VIA, VIA/VILI, *care*HPV, and cytology are referred to as screening tests while colposcopy and 4Q biopsy and followed by histology are termed confirmatory tests. Whilst the costs of all aforementioned tests are reported here, only the screening tests are included in the cost-effectiveness analysis.

### Diagnostic accuracy of screening strategies

CIN2+ prevalence was calculated based on diagnosis of CIN2+ from the study endpoint classification consensus by a panel of five pathologists in the HARP study [29]. Test performance was evaluated among the women recruited into the HARP study using local diagnostic outcomes as the reference test in calculation of the diagnostic accuracy of the index tests (VIA, VILI, cytology, HPV). Since the *care*HPV tests did not arrive in time for the study start, the diagnostic accuracy was taken from the HC-II HPV DNA test (also manufactured by QIAGEN Inc, Gaithesburg, MD). Excellent agreement (94.6%) between these tests was observed in the HARP population [30].

Screening strategies were defined as any standalone screening test or two screening tests conducted in series (triage options). For triage options, the second test is only conducted amongst those who test positive for the first test. The joint sensitivity of both tests within a strategy is reported for all participants screened, which increases the overall specificity of the triage option as compared with individual test accuracies or multiple tests in parallel. The number of cases detected was calculated by multiplying the joint sensitivity of the screening test(s) by the prevalence of CIN2+ in the population.

### Cost data

The cost data were collected during a one-month period in June 2012 and a one-week period in April 2013 in the HIV outpatient clinic and pathology laboratory in CHU-YO and from the CERBA laboratory in Ouagadougou. The cost analysis adopted a health care provider perspective (i.e. the facility providing the screening and care) [31]. An ingredients-based costing methodology was used, where quantities of resources were multiplied by their respective unit prices to obtain total costs per woman screened. Information was collected by direct observation of capital (equipment) and recurrent (personnel and consumables) resources used for each screening and diagnostic procedure. Resource use was collected only for activities directly related to the procedures, and costs related to the study were excluded. The time was taken from the mean of 10 women for each diagnostic and confirmatory test using a stopwatch. For the timings of the clinical procedures needed for each test, client preparation time and time spent on conducting the actual procedure were recorded separately to allow for disaggregation of the different screening procedures. The costs of VIA and VILI were collected separately and then grouped as a single procedure for combined VIA/VILI exam. Laboratory procedures were also observed for 10 procedures except for *care*HPV (when the test became available), which was only observed three times due to the need to do batch testing of 88 samples. The collection and analysis of careHPV test results in HARP has been previously described [23].

For the personnel costs, gross salaries were obtained from local salary data from the Ministry of Health and included basic salary, housing allowance, night shift allowance, risk allowance and responsibility analysis for personnel costs. The average cost per hour of working time

was obtained by dividing the annual salary of the staff member by the number of working hours (40 hours per week with 30 days annual leave and 14 official holidays) [32]. The average time spent by each staff member for each screening test per woman was multiplied by the average cost per minute of working time per staff member. Capital costs were annualised based on using local life expectancies of 2 to 10 years for equipment at the standard discount rate of 3% [33] and divided by the annual throughputs. Overhead costs related to all cervical cancer screening and confirmatory tests were applied at 25% of the total capital and recurrent costs [12]; this assumption was varied in the sensitivity analysis.

Costs do not include Value Added Tax as medical programmes and associated goods are exempt from it in Burkina Faso [34]. Whenever prices were obtained in another currency, the annual exchange rate for 2012 was used to convert into West African Francs (XOF) to get the in country price [35]. Costs are reported in United States Dollars (US$) using the average exchange rate for 2012 (US$ 1 = XOF 503.1) [35], and then inflated to 2019 US$ [36]. Credible ranges were not calculated for the costs.

Due to using the joint sensitivity of triage options, the total costs will vary depending on the number of tests given for the second screening test while the cases detected will be limited by the test with the lower sensitivity. For strategies requiring two laboratory test components, it was assumed that samples for both tests would be taken during the first clinical visit and that the sample for the second test would be analysed only if the first test was positive. For triage options with VIA/VILI, only options that gave the laboratory test second were examined as the cost of increased laboratory work for the non-VIA/VILI test and cost of an additional clinic visit cost guaranteed that giving a laboratory test first would be more expensive and therefore dominated in the cost-effectiveness analysis.

Routine practice for obtaining biopsies is to collect directly from a lesion seen during colposcopy (i.e. directed biopsy). Accordingly, the cost of a directed biopsy was calculated to compare with the systematic 4Q biopsy used under the study protocol, which may be advocated because of its potential increased sensitivity. The directed biopsy costs were calculated using assumptions based on interviews with study and site staff. Patient preparation, colposcopy and post-procedure time remained the same as for 4Q biopsy. For clinic staff time, the procedure time for the biopsy was divided by four to get the time for one biopsy. For laboratory staff time and consumables, the time was divided by two because the four samples were processed on two slides. Clinical consumables remained the same apart from only needing one container for the biopsy sample instead of two. Equipment costs remained the same.

## Cost-effectiveness analysis

Screening strategies were given to a hypothetical cohort of 1000 WLHIV over a time horizon from screening to diagnosis of CIN2+, which would range from the day of testing for VIA/VILI to months after taking the test for *care*HPV. Costs and outcomes were not discounted. Under the active follow up of the HARP study, the time from screening to CIN2+ diagnosis ranged from one to nine months, but this may take longer in reality. For triage options, all women received the first screening test but only the proportion who tested positive for the screening test received the second test so that:

$$C_{woman\ screened} = C_{s1} * N_{s1} + C_{s2} * N_{s2}$$

where $C_{s1}$ is the cost of the first screening test, $N_{s1}$ is the cohort size, $C_{s2}$ is the cost of the second screening test, and $N_{s2}$ is the percentage of women requiring the confirmatory test multiplied by the cohort size (i.e. those testing positive for the screening test). Since the sample for the second screening test was taken at the time of first test, no loss to follow up was assumed.

While the costs of confirmatory tests were not included in the cost-effectiveness analysis, the percentage of women that would need a colposcopy and the associated additional cost per cohort was reported alongside the number without CIN2+ receiving a colposcopy because of a positive screening test or triage strategy. In this way, the costs of false positives were incorporated into the second test of triage strategies in the cost-effectiveness analysis, and the costs of false negatives are not incorporated.

When comparing the options, screening strategies were ranked by cost and those that were dominated because they cost more and found fewer cases were removed as options. For those remaining screening strategies, incremental cost-effectiveness ratios (ICERs) were calculated for the cohort using the following formula:

$$ICER = \frac{Cost_2 - Cost_1}{Cases_2 - Cases_1}$$

Where *Cost* is the total cost of a screening strategy and *Cases* is the total number of CIN2 + cases detected for the corresponding strategy. For the first ICER calculated, the least costly option as the base case, and screening strategies that had cost-effectiveness ratios higher than the next, more effective, alternative were also removed due to extended dominance. While this analysis provides information on the cost per true case CIN2+ identified, it is limited in that it does not attempt to capture the long-term health impact of detecting a case, such as cases averted or life years saved.

## Sensitivity analyses

Parameters key to the incremental cost per additional CIN2+ case detected were varied in a one-way sensitivity analysis to investigate their impact on the results. The first analysis on the costs involved varying the flat rate on the overheads to 10% to get a low cost scenario and to 75% to approximate a high cost scenario [12]. In addition, the sensitivity of the screening strategies was increased and decreased by 20%.

As *care*HPV has not been widely introduced in SSA, detailed costings are reported here and a univariate sensitivity analysis was done on parameters to assess the impact of alternative assumptions on *care*HPV visit costs. The components of the *care*HPV test that were supplied by Qiagen included the *care*HPV brush, collection medium, test kit and machine; these were examined separately in the sensitivity analysis. Parameters examined in the univariate sensitivity analysis included: *care*HPV test component costs, consumable costs (excluding *care*HPV test components), equipment costs (excluding *care*HPV test components), staff costs, the number of women screened per clinic per year, the number of specimens per *care*HPV kit used, and the addition of laboratory training for *care*HPV. For VIA/VILI and cytology, a univariate analysis examined the impact of increasing and decreasing the costs by 20%. Finally, best and worst case scenarios were created using the lowest and highest values for all parameters for each test.

A threshold analysis was conducted on the cost of the *care*HPV screening strategies to see at what cost a screening strategy involving *care*HPV would become as cost-effective as the next best option. The total cost of items that were in addition to the *care*HPV test components is subtracted from this cost to find the price that the test components from Qiagen would need to be to get the strategy to be the most cost-effective option.

## Results

In total, 615 WLHIV were included in the HARP cohort in Burkina Faso, of whom 554 had histology results. The number of women with available screening strategy results ranged from

526 to 553. The median age was 36 years (inter-quartile range [IQR], 31–41). The prevalence of CIN2+ was 5.8% (95% confidence interval [CI]: 4.0–8.1).

Table 1 shows the costs and clinical time required per screening or confirmatory test. Screening tests were done in clinic by a nurse/midwife while confirmatory tests involved a gynaecologist assisted by a midwife. The average clinical time for each screening procedure was around 8 minutes with 95% CIs ranging from six to 10 minutes. The confirmatory tests took longer with a range from 13 minutes for colposcopy to 22 minutes for colposcopy with 4Q biopsy. Fig 1 shows the total costs per screening or confirmatory test. While cost of the screening tests ranged from US$3.2 for VIA to US$24.8 for cytology, confirmatory test costs ranged from US$6.6 for colposcopy without biopsy to US$48.0 with 4Q biopsy. Colposcopy with directed biopsy was US$33.0. The cost of 4Q biopsy was higher than directed biopsy mainly due to increases in pathology costs. The cost of each test broken down by components are shown in S1 Table.

Table 2 shows the cost per woman screened and incremental cost per CIN2+ case detected. The sensitivity of standalone test strategies ranged from 30% for HSIL cytology to 97% for *care*HPV. For the triage strategies, the joint sensitivity ranged from 23% for VIA/VILI followed by HSIL cytology to 70% for the combinations of LSIL cytology and *care*HPV. The range of costs per woman screened was US$3–34 while *care*HPV was US$23. The sensitivity of VIA/VILI was 56%. Adding *care*HPV to VIA/VILI increased the costs per woman screened by US$5 compared to VIA/VILI alone but did not change the sensitivity (56%). With an ICER of US$48 (range: US$37–60) compared with VIA (the least costly option), VIA/VILI was one of the two screening strategies that was not dominated along with *care*HPV (ICER = US$382, range: US$638–1014). It is important to note that while *care*HPV was the most expensive non-dominated option; it was also the screening strategy with the highest sensitivity (97%). Increasing the cost-effectiveness of these strategies would depend on the willingness to pay threshold per case of CIN2+ detected. None of the triage strategies were cost-effective as compared to single screening test strategies. This is because the increased cost of the second screening test compounded by finding fewer cases (or the same number of cases) due to the joint sensitivity. The one-way sensitivity analysis on the sensitivity of the screening options did not change which options were cost-effective (S2 Table).

**Table 1. Clinical timings and clinical and laboratory costs per person tested.**

| Type of procedure | Mean clinical time (minutes) | Total clinical costs | Total laboratory costs | Total costs (-/+20%) |
|---|---|---|---|---|
| *Screening Tests* | | | | |
| VIA | 7.8 (5.9 to 12.2) | $3.2 (3.1–3.5) | $0.0 | $3.2 (2.6–3.9) |
| VIA/VILI | 8.6 (6.4 to 13.6) | $3.6 (3.4–3.9) | $0.0 | $3.5 (2.9–4.3) |
| Cytology | 8.5 (6.5 to 11.9) | $6.6 (6.4–6.8) | $18.3 | $24.8 (19.8–29.8) |
| *care*HPV | 8.4 (6.2 to 11.7) | $5.5 (5.4–5.7) | $21.8 | $27.3 (21.8–32.8) |
| *Confirmatory tests* | | | | |
| Colposcopy | 13.1 (10.0 to 14.9) | $6.6 (6.4–6.7) | $0.0 | $6.6 (5.3–7.9) |
| Colposcopy and four-quadrant biopsy[2] | 21.8 (16.9 to 25.7) | $11.8 (11.5–12.0) | $36.1 | $47.9 (38.4–57.5) |
| Colposcopy and directed biopsy[2,3] | 16.7 (13.1 to 18.6) | $10.6 (10.3–10.7) | $22.4 | $33.0 (26.4–39.6) |

[1]Times are for midwife. Midwife times were slightly higher than gynaecologist times due to preparation responsibilities.

[2]The directed biopsy times and costs are estimated by interview and were not directly observed. Assumes that only one biopsy was taken.

Abbreviations: *care*HPV, HPV DNA test; VIA = visual inspection with acetic acid; VIA/VILI, combined visual inspection with Lugol's iodine.

All costs are in United States dollars (2019). Ranges for clinical time and clinical costs use the minimum and maximum values for clinical time. The range for the total costs is -/+20% of the base case estimate.

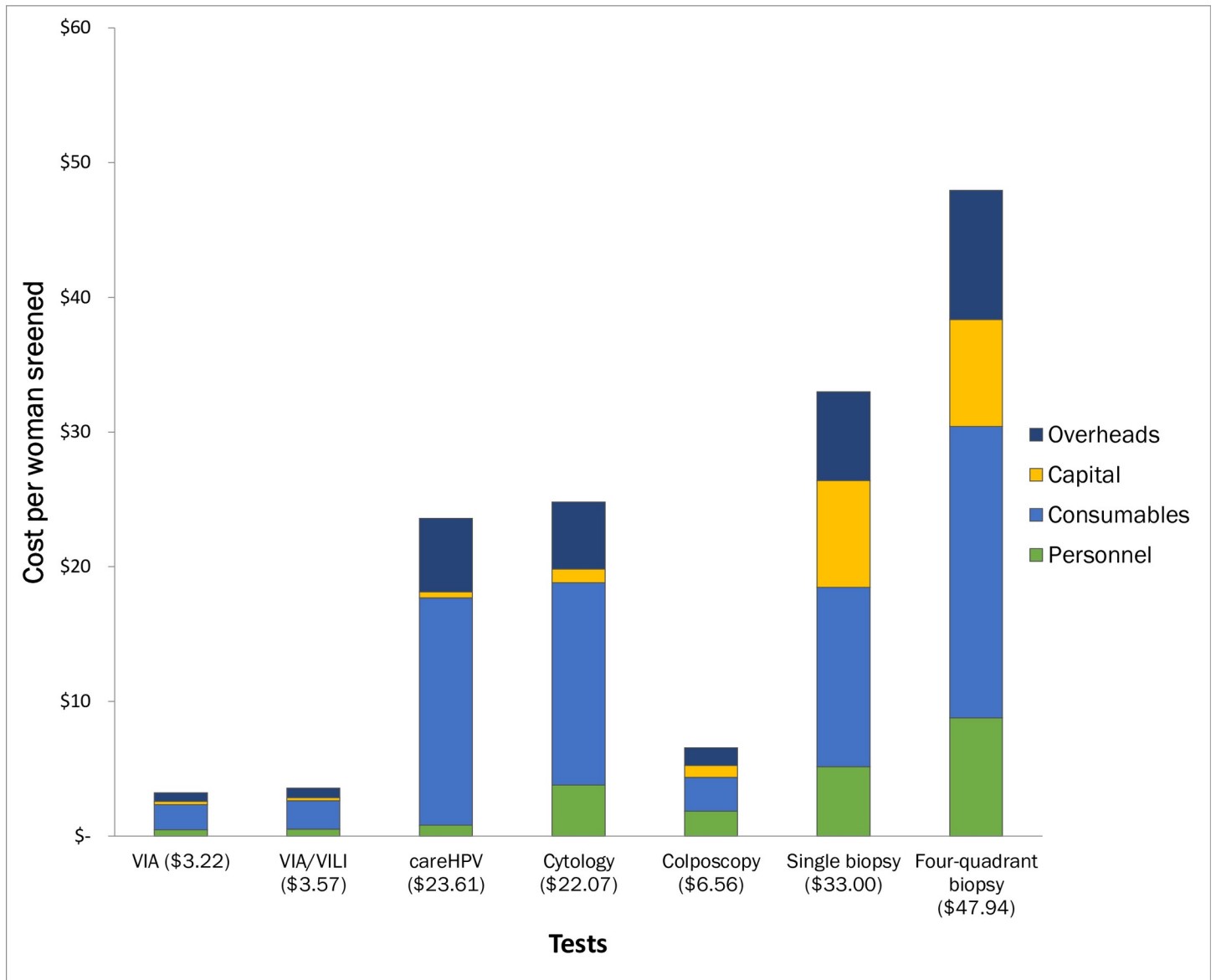

**Fig 1. Unit costs of screening and confirmatory tests in 2019 United States Dollars.**

Table 3 shows the main input parameters for screening via *care*HPV, and Fig 2 shows the results of the sensitivity analysis on these. The univariate analysis showed that the *care*HPV test cost had the highest impact on screening costs followed by laboratory staff training costs. The worst-case analysis for *care*HPV showed the costs increasing to US$42.1 per woman, just over twice the base case scenario.

The threshold analysis showed that *care*HPV would only become as cost-effective as VIA/VILI (i.e. have an ICER of US$48) if *care*HPV could be implemented without incurring lab training costs and if test components (machine, kit, sample medium and brush) were provided free of charge. Since it is unlikely that *care*HPV could be implemented without any of those costs, the threshold analysis indicates that cost per true case detected would certainly increase when using *care*HPV as compared to VIA/VILI. It is important to note, however, that this

**Table 2. Cost and cost-effectiveness of the screening strategies for a cohort of 1000 women in 2012 United States Dollars.**

| TEST OPTION | Sensitivity | Specificity | Positive predictive value | Cost per woman screened | Number of true CIN2 + cases detected | Incremental cost-effectiveness ratio | | | Percent (N) of women in the cohort receiving colposcopy | Percent (N) of women without CIN2 + receiving colposcopy | Additional cost of colposcopy per cohort |
|---|---|---|---|---|---|---|---|---|---|---|---|
| | | | | | | Base case | Low cost | High cost | | | |
| VIA | 44% | 80% | 13% | $3.2 | 25 | | | | 21% (210) | 18% (184) | $1,377 |
| VIA/VILI | 56% | 78% | 8% | $3.6 | 32 | $48 | $42 | $67 | 24% (239) | 21% (206) | $1,567 |
| VIA/VILI + *care*HPV | 56% | 89% | 64% | $8.2 | 32 | (dominated) | (dominated) | (dominated) | 14% (135) | 10% (102) | $888 |
| VIA/VILI + HSIL cytology | 23% | 99% | 10% | $8.6 | 13 | (dominated) | (dominated) | (dominated) | 2% (24) | 1% (11) | $161 |
| VIA/VILI + LSIL cytology | 40% | 93% | 55% | $8.6 | 23 | (dominated) | (dominated) | (dominated) | 9% (87) | 6% (64) | $569 |
| *care*HPV | 97% | 62% | 48% | $22.7 | 56 | $814 | $717 | $1,140 | 42% (418) | 36% (361) | $2,744 |
| HSIL cytology | 30% | 97% | 14% | $24.8 | 17 | (dominated) | (dominated) | (dominated) | 5% (45) | 3% (28) | $296 |
| LSIL cytology | 73% | 80% | 79% | $24.8 | 42 | (dominated) | (dominated) | (dominated) | 23% (226) | 18% (184) | $1,481 |
| HSIL cytology + *care*HPV | 30% | 98% | 13% | $28.1 | 17 | (dominated) | (dominated) | (dominated) | 4% (36) | 2% (19) | $237 |
| LSIL cytology+ *care*HPV | 70% | 88% | 95% | $31.2 | 40 | (dominated) | (dominated) | (dominated) | 16% (158) | 12% (118) | $1,036 |
| *care*HPV + HSIL cytology | 30% | 98% | 90% | $34.1 | 17 | (dominated) | (dominated) | (dominated) | 4% (36) | 2% (19) | $237 |
| *care*HPV + LSIL cytology | 70% | 88% | 95% | $34.1 | 40 | (dominated) | (dominated) | (dominated) | 16% (158) | 12% (118) | $1,036 |

Abbreviations: *care*HPV, HPV DNA test; CIN2+, high-grade cervical intraepithelial neoplasia; HSIL, high-grade squamous intraepithelial lesions; LSIL, low-grade squamous intraepithelial lesions; VIA, visual inspection with acetic acid; VIA/VILI, combined visual inspection with Lugol's iodine.

For triage options, the sensitivity, specificity, and positive predictive values are joint for both tests. The low cost incremental cost-effectiveness ratio uses 10% for overheads while the high cost uses 75% for overheads[a].

[a] For the cytology tests, this is the sensitivity of cytology to detect HSIL or LSIL.

switch would also increase the percentage of true cases found from 56% to 97%, potentially reducing the long-term healthcare costs for these women.

## Discussion

To our knowledge, this is the first study to report the cost-effectiveness for a wide range of cervical cancer screening strategies among WLHIV in Burkina Faso. These included currently recommended strategies in low-resource settings (VIA or VIA/VILI), cytology, and the HPV DNA test in the form of a rapid test (*care*HPV). Using cost data collected alongside a large evaluation study of screening approaches with systematic and rigorous histological endpoint determination, our analysis demonstrated a wide range in both the costs (US$3–34 per woman screened) and the sensitivity of the screening strategies (23–98%) to detect CIN2+. Testing costs were similar to the costs of integrating cervical cancer screening into HIV clinics in Kenya [24] and generally lower than those reported in WLHIV in South Africa [26]. The

**Table 3. Selected input parameters for *care*HPV in 2019 United States Dollars.**

| Parameter | Input | Assumptions |
|---|---|---|
| *care*HPV test costs | | |
| *care*HPV cervical sample brush | $0.56 | Quote from Qiagen invoice for HARP. |
| *care*HPV collection medium | $1.12 | |
| *care*HPV kit (96-well) | $1,079 | |
| *care*HPV machine | $22,484 | |
| Lifetime of *care*HPV machine | 5 years | Local practice for similar equipment. |
| Time needed for laboratory technician to process one 96-well kit | 4 hours & 3 minutes | Average of three observations. This includes hands off time. |
| Number of samples processed in 96-well kit | 88 | The test requires at least 4 controls per kit and an additional 2 blanks interspaced in panel to control for possible contamination (our practice, not manufacturer recommendation) and it was assumed that not every batch would be completely full. |
| Monthly salary | | |
| Midwife | $525 | Assumes that one third are Certified Midwife 1 and two thirds are Certified Midwife 2&3. |
| Lab technician | $572 | Assumes that half are laboratory technicians and half are senior laboratory technicians. |
| Sample carrier | $219 | Assumes that it takes 2 minutes to transport from clinic to on site laboratory and handover specimen. |

South African study showed that even when HPV DNA lab test costs were reduced to US$1 (2013), it would not be cost-effective as compared to the Papanicolaou test due to the lower specificity of HPV DNA testing compared to cervical cytology, which is standard care in this setting and performs with high sensitivity and specificity ensured by established infrastructure, expertise and external quality assessment programme in place [26].

No organised national screening programme or integration of cervical cancer screening into HIV care was in existence in Burkina Faso at the time of the study [11], and women often presented at late stages of cervical cancer and that were diagnosed clinically. Further barriers to screening in Burkina Faso included educational and socioeconomic barriers [37]. In target populations for cervical cancer screening such as WLHIV, the higher prevalence of cervical precancer would drive down the costs, thereby increasing the cost-effectiveness. Our results showed that the addition of VILI to VIA, the current screening modality in Burkina Faso increased sensitivity for CIN2+ from 44% to 56%, adding evidence to that provided by Muwonge *et al.* from Burkina Faso and other countries that demonstrated that VIA/VILI has a higher sensitivity than VIA alone [38]. While this would be considered too low for an effective screening strategy, the additional cost of VILI was only US$0.4 per woman screened, and the sensitivity was higher for CIN3+ (85%) [39]. In Burkina Faso, the HIV prevalence is now less than 1% [40]. In a study of sex workers in Bobo Dioulasso, Burkina Faso, 12% of WLHIV had HSIL lesions versus 1% in those who were HIV-seronegative [41]. If this trend follows to the rest of the population, then targeting WLHIV in HIV clinics provides an opportunity to detect and manage CIN2+ cases in order to prevent cervical cancer in a small proportion of the population who is at a much higher risk in a cost-effective way.

Recent WHO guidelines recommend HPV DNA triaged by visual inspection and a stand-alone HPV DNA test followed by treatment for low and middle-income countries [27]. In any triage combination involving *care*HPV and VIA/VILI, our results indicate that visual inspection followed by *care*HPV would have a lower cost compared to *care*HPV followed by visual inspection, due to the lower unit cost of visual inspection. *care*HPV cost US$19 more per

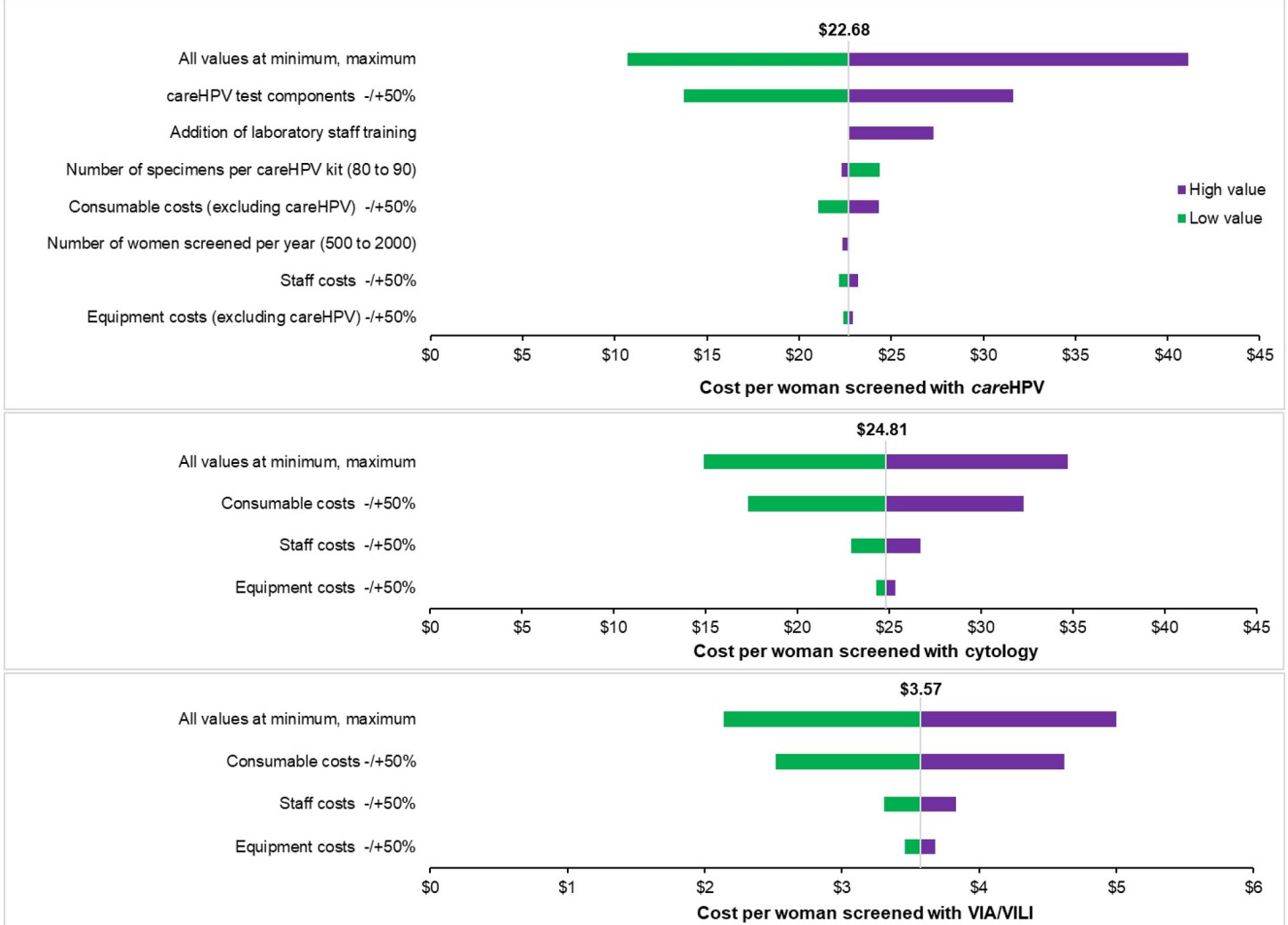

**Fig 2. One-way sensitivity analysis of the cost per woman screened with *care*HPV, cytology and visual inspection with acetic acid and Lugol's iodine (VIA/VILI) in 2019 United States Dollars (US$).**

woman screened than visual inspection. As both tests could be combined in one visit, this would result in a lower risk of loss to follow-up. In our study, the addition of VIA/VILI to *care*HPV resulted in a large decrease in sensitivity from 97% using *care*HPV alone to 56%, similar to that reported by others [22]. Frequent training and quality control for VIA/VILI may improve its diagnostic accuracy. In our study, nurses received training at study initiation only. Visual inspection may also have greater sensitivity to detect CIN3+ due to the larger lesions observed. When comparing standalone tests, VIA/VILI cost less per case detected than *care*HPV but missed nearly half (44%) of CIN2+. Increasingly the evidence indicates that the diagnostic accuracy of VIA/VILI is highly variable, although visual inspection enhanced with automated evaluation approaches report greater accuracy for CIN2+ [42,43]. While our analysis indicates that VIA/VILI is cost-effective compared to VIA alone and *care*HPV, it is hard to ignore the capability of DNA testing to detect 97% of cases.

Our study has a number of limitations. Firstly, due to issues with sourcing the tests in the timeframe needed for this study, the diagnostic accuracy results of *care*HPV were

derived from the performance of *Digene* HC-II in the HARP population; however, the diagnostic accuracy for *care*HPV and *Digene* HC-II tests was similar in a subset of HARP participants [30]. Accordingly, the costs for *care*HPV were used with the diagnostic accuracy from the *Digene* HC-II test, and differences in diagnostic accuracy could impact the ICER. Secondly, the costs of confirmatory tests, treatment and losses-to-follow-up were not included. The overall costs of confirmatory tests would be dependent on the specificity of the screening strategy but would result in a decrease in over-treatment. Loss to follow up circumvents the health gains that screening provides, and a lifetime model would be needed to capture the costs and health outcomes associated with it. While treatment increases the costs, the advantages of expedited diagnosis and treatment are recognised, including among WLHIV in South Africa [44], but are not captured in our analysis. In this study setting where the detection and treatment of CIN2+ were paid for by the study, none of the women refused colposcopy and only one did not receive the proposed treatment. In Burkina Faso, where the costs for procedures range from US$10 for cryotherapy to US$200 for hysterectomy, few women will be able or willing to pay these fees. Accordingly, public sector funding for the improvement of cervical cancer care needs to be carefully considered before implementing a screening program. A third of the women in this study (31%) were not on ART at enrolment, so earlier ART may impact the effectiveness of screening, particularly as the diagnostic accuracy of screening increases with higher CD4 counts [22,45].

Finally, while cost per additional CIN2+ case detected is a useful measure for comparing various screening strategies with each other, it does not answer whether screening in HIV clinics is a cost-effective option as compared to other interventions to prevent cervical cancer or as compared to interventions for other diseases. A model of the lifetime costs and benefits of cervical cancer screening programs using the cost per disability-adjusted life-year averted would be more appropriate as it would capture the long-term costs and benefits whilst enabling comparison with competing health interventions for other diseases [46]. A lifetime model would also be able to account for differences in the frequency of screening tests and possibly how these need to differ in WLHIV as compared to the general population. Cost per CIN2+ case detected, however, is more informative than the cost per woman screened, which can be used as a unit cost in future cost-effectiveness analyses.

## Conclusions

This study reports the costs of different screening strategies and diagnostic confirmatory tests for cervical cancer in WLHIV in Burkina Faso and the cost-effectiveness of screening and triage strategies. Our analysis showed that VIA/VILI alone and *care*HPV alone are potentially cost-effective options for cervical cancer screening for WLHIV in Burkina Faso, depending on the willingness-to-pay for each additional case detected. Whilst *care*HPV cost US$814 more per true case CIN2+ detected, its sensitivity was 97% as compared to 56% for VIA/VILI.

## Supporting information

**S1 Table. Breakdown of tests by cost component in 2019 United States Dollars.**
(DOCX)

**S2 Table. Results of the one-way sensitivity analysis on the sensitivity of screening strategies.** For the non-dominated screening strategies, the sensitivity was decreased and increased by 20%.
(DOCX)

## Acknowledgments

The authors wish to thank Leontine Bonkoungou, Mariam Nonguierma and Celine Bambara for assisting with the data collection for the clinical procedures.

Contributing members of the HARP study group included: A. Chikandiwa, E. Cutler, S. Delany-Moretlwe, D. A. Lewis, M.P. Magooa, V. Maseko, P. Michelow, B. Muzah, T. Omar, A. Puren (Johannesburg, South Africa); F. Djigma, J. Drabo, O. Goumbri-Lompo, N. Meda, B. Sawadogo, J. Simporé, A. Yonli, S Zan (Ouagadougou, Burkina Faso); V. Costes, M.N. Didelot, S. Doutre, N. Leventoux, N. Nagot, J. Ngou, M. Segondy (Montpellier, France); and A. Devine, C. Gilham, L. Gibson, H. Kelly, R. Legood, P. Mayaud, H.A. Weiss (London, UK).

The HARP Study Group also wishes to thank its International Scientific Advisory Group (ISAG) constituting of Prof. C. Lacey (Chair, University of York, UK), Prof. Y. Qiao (Chinese Academy of Medical Sciences and Peking Union Medical College, Beijing, China), Prof. M. Chirenje (University of Harare, Zimbabwe) and Prof. S. de Sanjosé (Institut Catala d'Oncologia, Barcelona, Spain), and external advice provided by Prof. W. Prendiville, former President of the International Federation of Colposcopy (IFCPC) (Dublin, Ireland), and Dr. A Olaitan, Gynaecological consultant (UCL, London, UK).

## Author Contributions

**Conceptualization:** Angela Devine, Rosa Legood, Philippe Mayaud.

**Data curation:** Angela Devine, Alice Vahanian, Bernard Sawadogo, Souleymane Zan, Fadima Yaya Bocoum, Helen Kelly, Clare Gilham.

**Formal analysis:** Angela Devine.

**Funding acquisition:** Nicolas Nagot, Rosa Legood, Nicolas Meda, Philippe Mayaud.

**Investigation:** Angela Devine, Alice Vahanian, Bernard Sawadogo.

**Methodology:** Angela Devine.

**Project administration:** Bernard Sawadogo.

**Supervision:** Rosa Legood, Alec Miners, Philippe Mayaud.

**Visualization:** Angela Devine.

**Writing – original draft:** Angela Devine.

**Writing – review & editing:** Alice Vahanian, Bernard Sawadogo, Souleymane Zan, Fadima Yaya Bocoum, Helen Kelly, Clare Gilham, Nicolas Nagot, Jason J. Ong, Rosa Legood, Nicolas Meda, Alec Miners, Philippe Mayaud.

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
