## [Decision Letter · Decision Letter 0]

2 Jun 2020

PONE-D-20-04078

Costs and cost-effectiveness of cervical cancer screening strategies in women living with HIV in Burkina Faso: the HPV in Africa Research Partnership (HARP) study

PLOS ONE

Dear Dr. Devine,

Thank you for submitting your manuscript to PLOS ONE. After careful consideration, we feel that it has merit based on the Reviewer reports but does not fully meet PLOS ONE’s publication criteria as it currently stands.

Therefore, we invite you to submit a revised version of the manuscript that addresses the points raised during the review process. 

Note that it is very important to carefully respond and revise as needed to address each point.

A marked-up copy of your manuscript that highlights changes made to the original version. You should upload this as a separate file labeled 'Revised Manuscript with Track Changes'.An unmarked version of your revised paper without tracked changes. You should upload this as a separate file labeled 'Manuscript'.

We look forward to receiving your revised manuscript.

Kind regards,

Bruce A Larson

Academic Editor

PLOS ONE

Journal Requirements:

'Ethical approval was granted by the Ministry of Health in BF (no. 2012-12-089), the Witwatersrand University in SA (no. 110707), and the London School of Hygiene and Tropical Medicine (no. 7400). '

a. Please amend your current ethics statement to include the full names of the ethics committee/institutional review board(s) that approved your specific study.

'The authors have declared that no competing interests exist. The careHPV and Digene HC-II kits used in this project were obtained through the QIAGEN Corporation donation program to the London School of Hygiene & Tropical Medicine.'

a. Please confirm that this does not alter your adherence to all PLOS ONE policies on sharing data and materials, by including the following statement: "This does not alter our adherence to  PLOS ONE policies on sharing data and materials.” (as detailed online in our guide for authors http://journals.plos.org/plosone/s/competing-interests). 

If there are restrictions on sharing of data and/or materials, please state these. Please note that we cannot proceed with consideration of your article until this information has been declared.

Reviewers' comments:

Reviewer's Responses to Questions

**Comments to the Author**

1. Is the manuscript technically sound, and do the data support the conclusions?

Reviewer #1: Yes

Reviewer #2: Partly

2. Has the statistical analysis been performed appropriately and rigorously? 

Reviewer #1: Yes

Reviewer #2: No

3. Have the authors made all data underlying the findings in their manuscript fully available?

Reviewer #1: Yes

Reviewer #2: No

4. Is the manuscript presented in an intelligible fashion and written in standard English?

Reviewer #1: Yes

Reviewer #2: Yes

5. Review Comments to the Author

Reviewer #1: Congratulation to the authors for this manuscript.

The work is about cost evaluation for one round of different screening tests and the number of colposcopies to finalize the diagnosis, having as reference the VIA, already performed in the local routine. I believe it is important to highlight this scenario, as CINs and cervical cancer have long-term to progress and HSIL screening is carried out in successive rounds, each method may provide very different cost-effectiveness strategies.

Clarify or add some information:

Introduction: mention that the careHPV DNA test is a kind of rapid test; Do you have any official recommendations for screening in Burkina Faso? Any specific recommendations for HIV+ women?

Methods: what is the period of the screening? (costs was related to the year 2012-2013); Cite some information about the technique of collecting and carrying out the careHPV DNA test.

Results: Have you had any cases of cervical cancer detected or was excluded from the beginning of the HARP cohort?

Page 16 (Line 3): careHPV (ICER = US $ 724 ...) versus the value in the Table 2 (ICER / Base case = $ 382), what is correct?

Page 17 (Last paragraph): how was found the 18% increase in true cases by careHVP?

Discussion: Important to discuss the no inclusion of the confirmatory tests in the cost-effectiveness analysis can mitigate the result of increased costs caused by the HPV test since it generated many more colposcopies.

The sensitivity of HSIL cytology and cytology alone was low, not feasible to be applied on the site under the present conditions (confirming a problem of collection/fixation or reading).

Conclusion: VIA + VILI is the best strategy evaluated in one round for the scenario studied. The careHPV DNA test did not improve the cost assessment when associated with VIA + VILI (high ICER).

Reviewer #2: GENERAL:

- This paper estimates the costs of different screening tests for cervical cancer prevention in Burkina Faso in women living with HIV, a group highly vulnerable to develop of cervical pre-cancer and cancer. As you note, there is limited cost information in the literature on screening and treatment in HIV treatment centers, and this paper provides important information to the field in that realm.

- However, I have strong reservations about the cost-effectiveness analysis as it uses cost per case detected as the outcomes rather than cases of pre-cancer or invasive cancer averted. This approach can result in misleading conclusions about the value of each screening modality. I would suggest that the authors consider presenting this manuscript as a cost and resource analysis of screening modalities rather than a cost-effectiveness analysis based on cost per case detected. I would encourage the authors to expand their analysis from cost per case detected to a cost-effectiveness analysis incorporating health impact, as this information would be very timely and useful and more appropriate reflect the value of different screening modalities. Please refer to recent consensus guidelines that have been published to guide economic evaluations related to cervical cancer prevention (https://www.sciencedirect.com/science/article/pii/S2405852119300230?via%3Dihub).

INTRODUCTION:

- Check punctuation throughout for missing periods, extraneous commas, etc.

- Page 11: "In order to expand access to cervical cancer screening...such as care HPV." Not a full sentence. Please revise for clarity.

- Page 11: Suggest providing some explanation of why VIA performs better in in HIV-negative women vs. WLHIV. Please also provide estimates of sensitivity of careHPV since it is mentioned to perform well in both groups.

-Page 11-12: Suggest revising this paragraph for clarity as it is contradictory as written. For example, it first states that the systematic review found HPV testing to be more cost-effective than Pap, but then the final sentence indicates that HPV DNA testing was not cost-effective compared to PAP. I appreciated the authors' comment on the implementation limitations of CC/HIV integration, but this point then got lost in cost and CE results. It would better highlight the need for this study to first discuss the gaps in costing and CE literature and findings to date, and then discuss how this paper is addressing those gaps.

METHODS:

- Do any women in Burkina Faso receive confirmatory testing after screening? If so, this would be important to include in CEA as different tests would trigger more/fewer colpos, which in turn can lead to better detection/linkage to treatment. If not, please state this (or other) rationale for excluding colposcopy/biopsy from CEA.

- Diagnostic Accuracy Section:

a) From the intro, it sounded like there was not a cervical cancer screening program in Burkina Faso, so please provide rationale for why VIA/VILI chosen to reflect "real life performance of the screening?"

b) Is the outpatient clinic referred to here different than the Hopital de Jour HIV clinic? Please clarify whether the test performance is assessed from the HARP study population (described in first methods paragraph) or not.

c) Please clarify what is meant by "study endpoint classification consensus" as method for determining CIN2+ prevalence. That is, is prevalence estimated from the study or estimated for the total population by expert opinion using disease classification criteria? Please clarify whether final classification of CIN2+ is based on screening tests or gold standard colposcopy/biopsy.

d) What is the potential impact of using HPV DNA test performance for careHPV on incremental costs/case detected since HC2 performs better (even marginally) than careHPV.

- Cost Data Section:

a) Please define unit of analysis of the micro-costing exercise (cost per woman screened?).

b) Standard depreciation rate  I think you mean the discount rate here, since that is often valued at 3%. The depreciation rate is calculated based on the expected life of the equipment.

c) Consider presenting results in 2019 XOF/USD -- this would support decision-makers in using/interpreting your results more easily.

d) “For triage options with VIA/VILI…”, it sounds like this choice was made to not overestimate the cost of HPV DNA lab testing. However, it may be a stronger assumption to follow the cadence of clinical/WHO guidelines or local standards for triage testing, as this will more accurately reflect the true cost of the screening modality in practice. For example, with HPV DNA testing, the HPV test is typically provided first, followed by triage with VIA.

e) Were credible ranges calculated for costs? Please report.

CEA Methods:

a) Please define the time horizon and model structure

b) Suggest including % requiring confirmatory test into the formula

c) How is uptake incorporated (for all tests)?

d) How is loss to follow-up factored in between the first and second test?

e) Please provide your rationale for not including colposcopy/biopsy in the CEA. Differences in the sensitivity/specificity of each screening test can lead to varying rates of referral to colposcopy (whether warranted with TP or not with FP), impacting both case detection, costs, and health outcomes. While access to colpo/biopsy varies, it is often available in secondary/tertiary care centers and a proportion of women would be referred for confirmatory testing.

f) Please explain how false test results (false positives/false negatives) were incorporated into the analysis and impact on cost per true case detected vs cost per case detected.

g) Strongly caution that the results can be misleading without information on downstream health impact incorporated (e.g., cases averted).

Sensitivity analysis:

a) It seems like the only sensitivity analysis conducted on the costs were on overhead rates and careHPV costs. However, cost components may be different in a study setting and/or under observation (e.g., time for procedures) than in real life. As such, please consider assessing the cost components for all screening tests over plausible ranges (e.g., +/- base case) or credible ranges if calculated.

b) Given the wide range of test sensitivity/specificity you report for WLHIV on page 11 and constraints of assessing costs in a single facility, it would be prudent to at least conduct a one-way sensitivity analysis for each screening strategy to assess the impact of parameter uncertainty on CEA results.

RESULTS

- Table 1: Since you report a range of clinical times, please also report the associated range of personnel costs per test. Also, given that each screening test was observed/ only 10 times, suggest reporting min/max instead of 95% CIs.

- It would be very helpful if you are able to include full costing results by cost component in table 3 or a supplementary appendix so that we can fully understand assumptions, potential areas of cost variation, strengths/limitations, representativeness, etc. The transparency will also make these results more accessible to key stakeholders like MOH, procurement teams, facility administrators, etc. to be used in real-world decision-making.

- Table 2:

a) Please clarify in footnote of table what is meant by LSIL cytology / HSIL cytology (I assume this means sensitivity of cytology to detect HSIL vs LSIL, but not clear as written).

b) Please present the positive predictive value of each test/combination of tests and total number of true CIN2+ cases detected, along with specificity, so readers have full information on test performance.

c) Please include denominator information for all counts and percents in the table.

d) Colpo costs: Please clarify whether these results are per woman receiving colpo or per 1,000 women in the cohort.

- Page 22, lines 5-7: “None of the triage strategies were cost-effective…” - please state what these strategies were compared to (WTP or other test strategies).

- Table 3: Time needed for laboratory technician - please clarify what is included in this time (inclusive/exclusive of hands off/processing time). If inclusive of hands off time then this is misleading as we expect staff would be able to complete other tasks during that time.

- Please report results of sensitivity analysis for all screening modalities, not just careHPV.

- Threshold analysis: please report the monetary cost results of the threshold analysis. As written, it is not clear why lab training and test components were selected for removal from price instead of other cost components.

- What is the impact of joint parameter uncertainty on results?

Discussion

- Page 18 - what are the differences from this analysis and the South Africa analysis referenced that had lower costs?

- “Secondly, the costs of treatment and LTFU, which would increase the ICERs....” This is not necessarily true since the ICER is a ratio of costs to outcomes and not additive. I think what you’re trying to say here is that inclusion of costs of treatment would increase the total cost per woman screened, however, this is misleading since it does not incorporate impact on health outcomes. Without including downstream health impacts (ability to successfully identify and treat pre-cancerous lesions, prevent a cervical cancer case or death) and associated costs, the “value” of each screening test is not actually being examined in this analysis. Further, LTFU has an impact on health outcomes and not just costs, as screening modalities that can support same day screen+treat (which report lower LTFU) are more likely to link CIN2+ cases to treatment for pre-cancer. As such, these results need to be reported with an abundance of caution in order to help the reader understand the limitations inherent in reporting a cost per case detected rather than cost per case averted.

- “Finally, while cost per additional CIN2+ case detected is a useful measure for comparing various screening strategies with each other, it does not answer whether screening in HIV clinics is a cost-effective option.” This is an incredibly important point. Given that this is the case, please provide your rationale for calculating/presenting cost-effectiveness of a case detected. Isolated from health outcomes, this information may mislead decision-makers into selecting the least costly test or a test that has the lowest ratio of specificity to cost per screen, but doesn’t necessarily yield the best value for preventing cervical pre-cancer or cancer.

- “If combining careHPV with visual inspection, our results indicate that visual inspection should be followed by careHPV.” Please explain whether this recommendation is based on clinical management/test performance or driven by costs, as the WHO recommends careHPV followed by VIA/VILI for improved test performance.

6. PLOS authors have the option to publish the peer review history of their article (what does this mean?). If published, this will include your full peer review and any attached files.

Reviewer #1: Yes: Julio C Teixeira

Reviewer #2: No

---

## [Author Response · Author response to Decision Letter 0]

20 Oct 2020

Reviewer #1: Congratulation to the authors for this manuscript.

The work is about cost evaluation for one round of different screening tests and the number of colposcopies to finalize the diagnosis, having as reference the VIA, already performed in the local routine. I believe it is important to highlight this scenario, as CINs and cervical cancer have long-term to progress and HSIL screening is carried out in successive rounds, each method may provide very different cost-effectiveness strategies.

Clarify or add some information:

Introduction: mention that the careHPV DNA test is a kind of rapid test; 

Given that samples for careHPV testing would need to be sent to a lab and run in batches, it is unlikely that women would receive the test results immediately. Accordingly, we do not describe it as a rapid test.

Do you have any official recommendations for screening in Burkina Faso? Any specific recommendations for HIV+ women?

As stated in the conclusions, the cost-effective screening options for WLHIV are VIA/VILI and careHPV (line 490):

“Our analysis showed that VIA/VILI alone and careHPV alone are potentially cost-effective options for cervical cancer screening for WLHIV in Burkina Faso, depending on the willingness-to-pay for each additional case detected. Whilst careHPV cost US$814 more per true case CIN2+ detected, its sensitivity was 97% as compared to 56% for VIA/VILI.”

Methods: what is the period of the screening? (costs was related to the year 2012-2013); 

This has been added at line 136: 

“A cross-sectional study of cervical cancer screening strategies was conducted among WLHIV aged between 25 and 50 years who were enrolled from the Hopital de Jour HIV clinic of the Centre Hospitalier Universitaire Yalgado Ouédraogo (CHU-YO) between November 2011 and April 2012.”

Cite some information about the technique of collecting and carrying out the careHPV DNA test.

We now refer to another publication that details this at line 202: 

“The collection and analysis of careHPV test results in HARP has been previously described [28].”

Results: Have you had any cases of cervical cancer detected or was excluded from the beginning of the HARP cohort?

These were exclude, line 147: 

“Eligibility criteria were women aged 25-50 years who were HIV-1 seropositive, resident in the city, who had not had any treatment for cervical cancer or hysterectomy, who were not pregnant or less than 8 weeks postpartum.”

Page 16 (Line 3): careHPV (ICER = US $ 724 ...) versus the value in the Table 2 (ICER / Base case = $ 382), what is correct?

$724 was correct with an error in Table 2, which has now been amended. The costs have also been inflated to 2019 US$ in response to Reviewer 2’s comments, so the ICER is now US$814.

Page 17 (Last paragraph): how was found the 18% increase in true cases by careHVP?

This has been amended so that it clearly reflects the data in Table 2: 

“Whilst careHPV cost US$814 more per true case CIN2+ detected, its sensitivity was 97% as compared to 56% for VIA/VILI.

Discussion: Important to discuss the no inclusion of the confirmatory tests in the cost-effectiveness analysis can mitigate the result of increased costs caused by the HPV test since it generated many more colposcopies.

This has been added to the limitations paragraph, line 457: 

“The overall costs of confirmatory tests would be dependent on the specificity of the screening strategy but would result in a decrease in over-treatment.”

The sensitivity of HSIL cytology and cytology alone was low, not feasible to be applied on the site under the present conditions (confirming a problem of collection/fixation or reading).

Our sensitivity if using HSIL+ threshold was low. This is in alignment with other studies in WLHIV where cytology based programmes are not routine but ASCUS+/LSIL+, sensitivity and specificity were reasonably high. We do not believe that the viability of the samples was the issue. The HARP study conducted an EQA programme for cytology between the site in Burkina Faso and the University of Montpellier for cytology. This data is part of another analysis and close to being published.

Conclusion: VIA + VILI is the best strategy evaluated in one round for the scenario studied. The careHPV DNA test did not improve the cost assessment when associated with VIA + VILI (high ICER).

We have revised the conclusion to reflect various comments by both reviewers about the cost-effectiveness (line 488): 

“Our analysis showed that VIA/VILI alone and careHPV alone are potentially cost-effective options for cervical cancer screening for WLHIV in Burkina Faso, depending on the willingness-to-pay for each additional case detected. Whilst careHPV cost US$382 more per true case CIN2+ detected, its sensitivity was 97% as compared to 56% for VIA/VILI.”

Reviewer #2: GENERAL:

- This paper estimates the costs of different screening tests for cervical cancer prevention in Burkina Faso in women living with HIV, a group highly vulnerable to develop of cervical pre-cancer and cancer. As you note, there is limited cost information in the literature on screening and treatment in HIV treatment centers, and this paper provides important information to the field in that realm.

- However, I have strong reservations about the cost-effectiveness analysis as it uses cost per case detected as the outcomes rather than cases of pre-cancer or invasive cancer averted. This approach can result in misleading conclusions about the value of each screening modality. I would suggest that the authors consider presenting this manuscript as a cost and resource analysis of screening modalities rather than a cost-effectiveness analysis based on cost per case detected. I would encourage the authors to expand their analysis from cost per case detected to a cost-effectiveness analysis incorporating health impact, as this information would be very timely and useful and more appropriate reflect the value of different screening modalities. Please refer to recent consensus guidelines that have been published to guide economic evaluations related to cervical cancer prevention (https://www.sciencedirect.com/science/article/pii/S2405852119300230?via%3Dihub).

While we agree that a cost-effectiveness analysis using DALYs instead of cases detected would be preferable, we do not have the resources to build a lifetime model for cancer progression. We have considered switching to a cost analysis, but we believe that it would be more misleading as the focus would shift to the cost per woman screened rather than the cost per case detected. When looking at the costs alone, the cheapest option (VIA) also detects the least cases. With a cost per case detected, we are able to start looking at the value gained for the cost, even if it is not comparable with other interventions for cervical cancer prevention or other diseases.

In order to further emphasize the limitations of this approach, we have added further details to the discussion (line 462): 

“Finally, while cost per additional CIN2+ case detected is a useful measure for comparing various screening strategies with each other, it does not answer whether screening in HIV clinics is a cost-effective option as compared to other interventions to prevent cervical cancer or as compared to interventions for other diseases. A model of the lifetime costs and benefits of cervical cancer screening programs using the cost per disability-adjusted life-year averted would be more appropriate as it would capture the long-term costs and benefits whilst enabling comparison with competing health interventions for other diseases. A lifetime model would also be able to account for differences in the frequency of screening tests and possibly how these need to differ in WLHIV as compared to the general population. Cost per CIN2+ case detected, however, is more informative than the cost per woman screened, which can be used as a unit cost in future cost-effectiveness analyses.”

INTRODUCTION:

- Check punctuation throughout for missing periods, extraneous commas, etc.

This has been checked throughout the text, and we hope that these issues have been corrected.

- Page 11: "In order to expand access to cervical cancer screening...such as care HPV." Not a full sentence. Please revise for clarity.

Line 92 has been revised to the following: 

“In order to expand access to cervical cancer screening, less resource-intensive techniques are being employed, such as VIA/VILI that can be performed by trained nurses and midwives who can immediately treat lesions with cryotherapy (screen and treat approach).”

- Page 11: Suggest providing some explanation of why VIA performs better in in HIV-negative women vs. WLHIV. Please also provide estimates of sensitivity of careHPV since it is mentioned to perform well in both groups.

It is unclear whether VIA performs better in HIV negative women as compared to WLHIV. Other factors, such as training and operator experience, may explain the low sensitivity. We have expanded on this in the introduction (line 101): 

“Visual inspection has shown to have reasonable sensitivity and specificity for CIN2+/CIN3+ in the general population [17], but can be highly variable depending on setting and operator training and experience [18]. It is unclear if diagnostic accuracy of visual inspection is dependent on HIV status, as few studies evaluating diagnostic accuracy by HIV status report conflicting findings [19, 20]. The pooled sensitivity of VIA for CIN2+ in HIV-negative women in SSA has been shown to vary from 76% to 87% [17]. The careHPV DNA test has had satisfactorily high and equivalent sensitivity in various settings among HIV-seronegative women [21] and in WLHIV [22] but lower specificity to distinguish CIN2+ among WLHIV due to higher prevalence of HR-HPV in WLHIV [22, 23].” 

-Page 11-12: Suggest revising this paragraph for clarity as it is contradictory as written. For example, it first states that the systematic review found HPV testing to be more cost-effective than Pap, but then the final sentence indicates that HPV DNA testing was not cost-effective compared to PAP. I appreciated the authors' comment on the implementation limitations of CC/HIV integration, but this point then got lost in cost and CE results. It would better highlight the need for this study to first discuss the gaps in costing and CE literature and findings to date, and then discuss how this paper is addressing those gaps.

We have revised this paragraph so that it better highlights the gaps in the literature (line 113): 

“To date there has been limited costing evidence on cervical screening in SSA [9, 24, 25], and the cost-effectiveness of the careHPV test against histological outcomes has only been assessed in a handful of countries [14, 16, 26]. A recent systematic review of the cost-effectiveness of cervical cancer screening strategies in LMICs (not limited to WLHIV) indicated that visual inspection and HPV testing are more cost-effective than cytology and that the cost-effectiveness of HPV testing was dependent on the test costs and performance of visual inspection [9]. The frequent and intensive follow-up of patients at HIV clinics may provide a valuable opportunity to organise cervical cancer screening. Targeting WLHIV is a commonly recommended policy option in SSA [27]; however, this has been limited in implementation. Few studies have investigated the costs and cost-effectiveness of integrating cervical cancer screening into HIV services, and none of these studies have been conducted in Burkina Faso [24, 26]. For example, a cost study in Kenya indicated that integrating cervical screening through VIA, VIA/VILI, careHPV, Papanicolaou, or Hybrid Capture II (HC-II) into HIV services would be less costly than a non-integrated programme due to economies of scope (i.e. efficiencies due to broadening services offered) [24].”

METHODS:

- Do any women in Burkina Faso receive confirmatory testing after screening? If so, this would be important to include in CEA as different tests would trigger more/fewer colpos, which in turn can lead to better detection/linkage to treatment. If not, please state this (or other) rationale for excluding colposcopy/biopsy from CEA.

Women would typically be treated after screening. We now describe our reporting of the costs of confirmatory testing in line 264: 

“While the costs of confirmatory tests were not included in the cost-effectiveness analysis, the percentage of women that would need a colposcopy and the associated additional cost per cohort was reported alongside the number without CIN2+ receiving a colposcopy because of a positive screening test or triage strategy. In this way, the costs of false positives were incorporated into the second test of triage strategies in the cost-effectiveness analysis, and the costs of false negatives are not incorporated.”

- Diagnostic Accuracy Section:

a) From the intro, it sounded like there was not a cervical cancer screening program in Burkina Faso, so please provide rationale for why VIA/VILI chosen to reflect "real life performance of the screening?"

You are correct, there wasn’t a cervical cancer screening program at the time of the study. We have rephrased this paragraph, deleting the mention of real life performance.

b) Is the outpatient clinic referred to here different than the Hopital de Jour HIV clinic? Please clarify whether the test performance is assessed from the HARP study population (described in first methods paragraph) or not.

Yes, this is the same clinic. This has been edited for clarity (line 168): 

“Test performance was evaluated among the women recruited into the HARP study...”

c) Please clarify what is meant by "study endpoint classification consensus" as method for determining CIN2+ prevalence. That is, is prevalence estimated from the study or estimated for the total population by expert opinion using disease classification criteria? Please clarify whether final classification of CIN2+ is based on screening tests or gold standard colposcopy/biopsy.

We have added further details on the endpoint classification at line 166:

“CIN2+ prevalence was calculated based on diagnosis of CIN2+ from the study endpoint classification consensus by a panel of five pathologists in the HARP study [29].”

d) What is the potential impact of using HPV DNA test performance for careHPV on incremental costs/case detected since HC2 performs better (even marginally) than careHPV.

If HCII were to perform better than careHPV, then we would expect the ICER to increase. If the opposite were true, then we would expect it to decrease. We now explicitly state this in the discussion (line 454): 

“Accordingly, the costs for careHPV were used with the diagnostic accuracy from the Digene HC-II test, and differences in diagnostic accuracy could impact the ICER.”

We have also added details on the agreement in the methods (line 173): 

“Excellent agreement (94.6%) between these tests was observed in the HARP population [30].”

- Cost Data Section:

a) Please define unit of analysis of the micro-costing exercise (cost per woman screened?).

We have clarified that this is cost per woman screened in line 190: 

“An ingredients-based costing methodology was used where quantities of resources were multiplied by their respective unit prices to obtain total costs per woman screened.”

b) Standard depreciation rate  I think you mean the discount rate here, since that is often valued at 3%. The depreciation rate is calculated based on the expected life of the equipment.

This has been changed to discount rate in line 214:

“Capital costs were annualised based on using local life expectancies of 2 to 10 years for equipment at the standard discount rate of 3%”

c) Consider presenting results in 2019 XOF/USD -- this would support decision-makers in using/interpreting your results more easily.

Results have been updated to 2019 USD throughout.

d) “For triage options with VIA/VILI…”, it sounds like this choice was made to not overestimate the cost of HPV DNA lab testing. However, it may be a stronger assumption to follow the cadence of clinical/WHO guidelines or local standards for triage testing, as this will more accurately reflect the true cost of the screening modality in practice. For example, with HPV DNA testing, the HPV test is typically provided first, followed by triage with VIA.

This decision is explained and discussed in detail at line 419:

“Recent WHO guidelines recommend HPV DNA triaged by visual inspection and a standalone HPV DNA test followed by treatment for low and middle-income countries [27]. In any triage combination involving careHPV and VIA/VILI, our results indicate that visual inspection followed by careHPV would have a lower cost compared to careHPV followed by visual inspection, due to the lower unit cost of visual inspection. careHPV cost US$19 more per woman screened than visual inspection. As both tests could be combined in one visit, this would result in a lower risk of loss to follow-up. In our study, the addition of VIA/VILI to careHPV resulted in a large decrease in sensitivity from 97% using careHPV alone to 56%, similar to that reported by others [22]. Frequent training and quality control for VIA/VILI may improve its diagnostic accuracy. In our study, nurses received training at study initiation only. Visual inspection may also have greater sensitivity to detect CIN3+ due to the larger lesions observed. When comparing standalone tests, VIA/VILI costs less per case detected than careHPV but missed nearly half (44%) of CIN2+. Increasingly the evidence indicates that the diagnostic accuracy of VIA/VILI is highly variable, although visual inspection enhanced with automated evaluation approaches report greater accuracy for CIN2+ [42, 43]. While our analysis indicates that it is cost-effective compared to VIA alone, it is hard to ignore the capability of DNA testing to detect 97% of cases.”

We have added the following to the methods to further clarify this decision at line 232: 

“For triage options with VIA/VILI, only options that gave the laboratory test second were examined as the cost of increased laboratory work for the non-VIA/VILI test and cost of an additional clinic visit cost guaranteed that giving a laboratory test first would be more expensive and therefore dominated in during the cost-effectiveness analysis.”

e) Were credible ranges calculated for costs? Please report.

Credible ranges were not calculated. Line 225 now states this: 

“Credible ranges were not calculated for the costs.”

CEA Methods:

a) Please define the time horizon and model structure

The time horizon is from screening to diagnosis, which could vary depending on how long it would take to get enough samples to run the careHPV test. We now state in line 251: 

“Screening strategies were given to a hypothetical cohort of 1000 WLHIV over a time horizon from screening to diagnosis of CIN2+, which would range from the day of testing for VIA/VILI to approximately one month after taking the test for careHPV.” 

b) Suggest including % requiring confirmatory test into the formula

Again, since treatment would normally take place before confirmatory testing, this has not been explicitly included in our cost-effectiveness results.

c) How is uptake incorporated (for all tests)?

It was assumed that all strategies would have the same uptake. 

d) How is loss to follow-up factored in between the first and second test?

We did not explicitly include loss to follow-up in our results. This is now stated at line 262: 

“Since the sample for the second screening test was taken at the time of first test, no loss to follow up was assumed.”

e) Please provide your rationale for not including colposcopy/biopsy in the CEA. Differences in the sensitivity/specificity of each screening test can lead to varying rates of referral to colposcopy (whether warranted with TP or not with FP), impacting both case detection, costs, and health outcomes. While access to colpo/biopsy varies, it is often available in secondary/tertiary care centers and a proportion of women would be referred for confirmatory testing.

As described above, colposcopy/biopsy is not part of standard care and so is not included in the cost-effectiveness analysis. Table 2 reports the % of women who would require referral to colposcopy if it were to be used before treatment.

f) Please explain how false test results (false positives/false negatives) were incorporated into the analysis and impact on cost per true case detected vs cost per case detected.

We have added the following to the text (line 267):

“In this way, the cost of false positives are incorporated into the second test of triage strategies in the cost-effectiveness analysis, and the cost of false negatives are not incorporated.”

g) Strongly caution that the results can be misleading without information on downstream health impact incorporated (e.g., cases averted).

We have included a statement at the end of this section (line 264): 

“While this analysis provides information on the cost per true case CIN2+ identified, it is limited in that it does not attempt to capture the long-term health impact of detecting a case, such as cases averted or life years saved.” 

In conjunction with the revised discussion, the limitations of this study should be clear to the reader.

Sensitivity analysis:

a) It seems like the only sensitivity analysis conducted on the costs were on overhead rates and careHPV costs. However, cost components may be different in a study setting and/or under observation (e.g., time for procedures) than in real life. As such, please consider assessing the cost components for all screening tests over plausible ranges (e.g., +/- base case) or credible ranges if calculated.

Table 1 now reports total costs for each test with +/-20% of the base case value.

b) Given the wide range of test sensitivity/specificity you report for WLHIV on page 11 and constraints of assessing costs in a single facility, it would be prudent to at least conduct a one-way sensitivity analysis for each screening strategy to assess the impact of parameter uncertainty on CEA results.

This has been added in S2 Table.

RESULTS

- Table 1: Since you report a range of clinical times, please also report the associated range of personnel costs per test. Also, given that each screening test was observed/ only 10 times, suggest reporting min/max instead of 95% CIs.

Table 1 now has ranges with min/max instead of 95% CIs and the associated clinical cost time ranges have now been added.

- It would be very helpful if you are able to include full costing results by cost component in table 3 or a supplementary appendix so that we can fully understand assumptions, potential areas of cost variation, strengths/limitations, representativeness, etc. The transparency will also make these results more accessible to key stakeholders like MOH, procurement teams, facility administrators, etc. to be used in real-world decision-making.

Supplementary Table 1 has been added to provide details on cost components for each screening and confirmatory test.

- Table 2:

a) Please clarify in footnote of table what is meant by LSIL cytology / HSIL cytology (I assume this means sensitivity of cytology to detect HSIL vs LSIL, but not clear as written).

A footnote has been added to Table 2.

b) Please present the positive predictive value of each test/combination of tests and total number of true CIN2+ cases detected, along with specificity, so readers have full information on test performance.

The PPV and specificity have been added to Table 2, and we have clarified that cases detected are true cases.

c) Please include denominator information for all counts and percents in the table.

We now explicitly stated that this is for a cohort of 1000 women in the table legend.

d) Colpo costs: Please clarify whether these results are per woman receiving colpo or per 1,000 women in the cohort.

This is the cost per cohort and is now stated in the table.

- Page 22, lines 5-7: “None of the triage strategies were cost-effective…” - please state what these strategies were compared to (WTP or other test strategies).

As described in the methods, screening strategies that were dominated were removed from the analysis. We have clarified this in the text. Line 360: 

“None of the triage strategies were cost-effective as compared to other test strategies. This is because the increased cost of the second screening test compounded by finding fewer cases (or the same number of cases) due to the joint sensitivity.”

- Table 3: Time needed for laboratory technician - please clarify what is included in this time (inclusive/exclusive of hands off/processing time). If inclusive of hands off time then this is misleading as we expect staff would be able to complete other tasks during that time.

We now state in Table 3 that this includes hands off time. We chose to include this in order to provide a conservative estimate of costs, and the threshold analysis on careHPV helps to address this issue.

- Please report results of sensitivity analysis for all screening modalities, not just careHPV.

Figure 2 includes results of the sensitivity analyses for VIA/VILI and cytology. Note that further analyses were done on careHPV due to the nature of batch testing and uncertainties about what prices would be set for test components.

- Threshold analysis: please report the monetary cost results of the threshold analysis. As written, it is not clear why lab training and test components were selected for removal from price instead of other cost components.

As stated in the methods, line 308: 

“The total cost of the items that were not in addition to the careHPV test components is subtracted from this cost to find the price that the test components from Qiagen would need to be to get the strategy to be the most cost-effective option.” 

We have rephrased the results for clarity: “The threshold analysis showed that careHPV would only become as cost-effective as VIA/VILI (i.e. have an ICER of US$48) if careHPV could be implemented without incurring lab training costs and if test components (machine, kit, sample medium and brush) were provided free of charge. Since it is unlikely that careHPV could be implemented without any of those costs, the threshold analysis indicates that cost per true case detected would certainly increase when using careHPV as compared to VIA/VILI. It is important to note, however, that this switch would also increase the percentage of true cases found from 56% to 97%, potentially reducing the long-term healthcare costs for these women.”

- What is the impact of joint parameter uncertainty on results?

We are not able to answer this through our analysis.

Discussion

- Page 18 - what are the differences from this analysis and the South Africa analysis referenced that had lower costs?

We have added details on the South African study at line 403:

“The South African study showed that even when HPV DNA lab test costs were reduced to US$1 (2013), it would not be cost-effective as compared to the Papanicolaou test due to the lower specificity of HPV DNA testing compared to cervical cytology, which is standard care in this setting and performs with high sensitivity and specificity ensured by established infrastructure, expertise and external quality assessment programme in place [26].”

- “Secondly, the costs of treatment and LTFU, which would increase the ICERs....” This is not necessarily true since the ICER is a ratio of costs to outcomes and not additive. I think what you’re trying to say here is that inclusion of costs of treatment would increase the total cost per woman screened, however, this is misleading since it does not incorporate impact on health outcomes. Without including downstream health impacts (ability to successfully identify and treat pre-cancerous lesions, prevent a cervical cancer case or death) and associated costs, the “value” of each screening test is not actually being examined in this analysis. Further, LTFU has an impact on health outcomes and not just costs, as screening modalities that can support same day screen+treat (which report lower LTFU) are more likely to link CIN2+ cases to treatment for pre-cancer. As such, these results need to be reported with an abundance of caution in order to help the reader understand the limitations inherent in reporting a cost per case detected rather than cost per case averted.

We have revised the text so that the impacts of loss to follow up and treatment are clearer (line 456): “Secondly, the costs of confirmatory tests, treatment and losses-to-follow-up were not included. The overall costs of confirmatory tests would be dependent on the specificity of the screening strategy but would result in a decrease in over-treatment. Loss to follow up circumvents the health gains that screening provides, consequently increasing the costs. While treatment increases the costs, the advantages of expedited diagnosis and treatment are recognised, including among WLHIV in South Africa [44], but are not captured in our analysis.”

- “Finally, while cost per additional CIN2+ case detected is a useful measure for comparing various screening strategies with each other, it does not answer whether screening in HIV clinics is a cost-effective option.” This is an incredibly important point. Given that this is the case, please provide your rationale for calculating/presenting cost-effectiveness of a case detected. Isolated from health outcomes, this information may mislead decision-makers into selecting the least costly test or a test that has the lowest ratio of specificity to cost per screen, but doesn’t necessarily yield the best value for preventing cervical pre-cancer or cancer.

As detailed above, we have rewritten this paragraph to clarify these points (line 473):

“Finally, while cost per additional CIN2+ case detected is a useful measure for comparing various screening strategies with each other, it does not answer whether screening in HIV clinics is a cost-effective option as compared to other interventions to prevent cervical cancer or as compared to interventions for other diseases. A model of the lifetime costs and benefits of cervical cancer screening programs using the cost per disability-adjusted life-year averted would be more appropriate as it would capture the long-term costs and benefits whilst enabling comparison with competing health interventions for other diseases [46]. A lifetime model would also be able to account for differences in the frequency of screening tests and possibly how these need to differ in WLHIV as compared to the general population. Cost per CIN2+ case detected, however, is more informative than the cost per woman screened, which can be used as a unit cost in future cost-effectiveness analyses.”

- “If combining careHPV with visual inspection, our results indicate that visual inspection should be followed by careHPV.” Please explain whether this recommendation is based on clinical management/test performance or driven by costs, as the WHO recommends careHPV followed by VIA/VILI for improved test performance.

We have revised this section to clarify our argument (line 419):

“Recent WHO guidelines recommend HPV DNA triaged by visual inspection and a standalone HPV DNA test followed by treatment for low and middle-income countries [27]. In any triage combination involving careHPV and VIA/VILI, our results indicate that visual inspection followed by careHPV would have a lower cost compared to careHPV followed by visual inspection, due to the lower unit cost of visual inspection. careHPV cost US$19 more per woman screened than visual inspection. As both tests could be combined in one visit, this would result in a lower risk of loss to follow-up. In our study, the addition of VIA/VILI to careHPV resulted in a large decrease in sensitivity from 97% using careHPV alone to 56%, similar to that reported by others [22]. Frequent training and quality control for VIA/VILI may improve its diagnostic accuracy. In our study, nurses received training at study initiation only. Visual inspection may also have greater sensitivity to detect CIN3+ due to the larger lesions observed. When comparing standalone tests, VIA/VILI costs less per case detected than careHPV but missed nearly half (44%) of CIN2+. Increasingly the evidence indicates that the diagnostic accuracy of VIA/VILI is highly variable, although visual inspection enhanced with automated evaluation approaches report greater accuracy for CIN2+ [42, 43]. While our analysis indicates that it is cost-effective compared to VIA alone, it is hard to ignore the capability of DNA testing to detect 97% of cases.”

---

## [Decision Letter · Decision Letter 1]

23 Dec 2020

PONE-D-20-04078R1

Costs and cost-effectiveness of cervical cancer screening strategies in women living with HIV in Burkina Faso: the HPV in Africa Research Partnership (HARP) study

PLOS ONE

Dear Dr. Devine,

Thank you for submitting your manuscript to PLOS ONE. After careful consideration, we feel that it has merit but does not fully meet PLOS ONE’s publication criteria as it currently stands. Therefore, we invite you to submit a revised version of the manuscript that addresses the points raised during the review process.

We look forward to receiving your revised manuscript.

Kind regards,

Bruce A Larson

Academic Editor

PLOS ONE

Additional Editor Comments (if provided):

The revised manuscript is substantially improved and addressed the comments, suggestions, and recommendations of the reviewers.

Reviewer 2 provides a short, final, list to comments for you to address as you make a final revision of your manuscript. As long as you address these comments and revise accordingly, I as the Academic Editor and review. As long as your revise and explain in your response letter how you addressed each (and I think the response is clear), I suspect we will not need to send again the reviewers.

Reviewers' comments:

Reviewer's Responses to Questions

**Comments to the Author**

1. If the authors have adequately addressed your comments raised in a previous round of review and you feel that this manuscript is now acceptable for publication, you may indicate that here to bypass the “Comments to the Author” section, enter your conflict of interest statement in the “Confidential to Editor” section, and submit your "Accept" recommendation.

Reviewer #1: All comments have been addressed

Reviewer #2: (No Response)

2. Is the manuscript technically sound, and do the data support the conclusions?

Reviewer #1: (No Response)

Reviewer #2: Yes

3. Has the statistical analysis been performed appropriately and rigorously? 

Reviewer #1: (No Response)

Reviewer #2: Yes

4. Have the authors made all data underlying the findings in their manuscript fully available?

Reviewer #1: (No Response)

Reviewer #2: Yes

5. Is the manuscript presented in an intelligible fashion and written in standard English?

Reviewer #1: (No Response)

Reviewer #2: Yes

6. Review Comments to the Author

Reviewer #1: (No Response)

Reviewer #2: The authors have adequately addressed most of the questions raised in the previous review and the paper is much stronger. However, a few minor comments are included below that would further improve clarity and better interpretation of the results.

1. Thank you for the updates to the manuscript and particularly the limitations/discussion section on the challenges associated with interpreting cost per case detected, since conclusions and recommendations may be different when evaluating cervical cancer prevention as the end point. Please consider including this in the abstract conclusions as well, to support appropriate reader (and potential end-user) interpretation of cost per case detected results.

2. Table 2. The additional details in the manuscript and Table 2 (PPV, true CIN2+ cases detected and colposcopy) are helpful. However, the information on colposcopy in Table 2 is a bit laborious for the reader to interpret as presented (some percents, some #s). Consider including both Ns and %s for the columns of women receiving colpo and women without CIN2 receiving colpo, or clarify interpretation of these columns with a table footnote to improve clarity.

3. Thank you for noting that a sensitivity analysis for VIA/VILI and cytology were conducted. The author response suggests this is included in Figure 2, but Figure 2 appears to be reporting the univariate analysis for careHPV. Please clarify where this information can be found.

4. Time horizon: thanks for adding the time horizon. Suggest explicitly noting whether costs/outcomes were discounted (assuming not because the time horizon was less than one year).

5. Line 446-448: "While our analysis indicates that it is cost-effective compared to VIA alone, it is hard to ignore the capability of DNA testing to detect 97% of cases.” Not clear what "it is cost-effective compared to VIA alone" is referring to. If careHPV, should it be "less cost-effective compared to VIA alone" to align with the results? Please clarify.

6. Line 459-460: "Loss to follow up circumvents the health gains that screening provides, consequently increasing the costs." Agree with the first part of this statement, however, LTFU may not necessarily result in increased costs when looking across the lifetime and evaluating cervical cancer cases averted (i.e., increased linkage to care often = higher costs associated with treatment). Suggest revising for clarity.

7. PLOS authors have the option to publish the peer review history of their article (what does this mean?). If published, this will include your full peer review and any attached files.

Reviewer #1: **Yes: **Julio Cesar Teixeira

Reviewer #2: No

---

## [Author Response · Author response to Decision Letter 1]

27 Jan 2021

1. Thank you for the updates to the manuscript and particularly the limitations/discussion section on the challenges associated with interpreting cost per case detected, since conclusions and recommendations may be different when evaluating cervical cancer prevention as the end point. Please consider including this in the abstract conclusions as well, to support appropriate reader (and potential end-user) interpretation of cost per case detected results.

Thanks very much for this and the following helpful suggestions to further improve the manuscript. We have amended the abstract conclusions to (lines 59-64): 

“Depending on the willingness to pay for the detection of a case of cervical cancer, decision makers in Burkina Faso can consider a variety of cervical cancer screening strategies for WLHIV. While careHPV is more costly, it has the potential to be cost-effective depending on the willingness to pay threshold. Future research should explore the lifetime costs and benefits of cervical cancer screening to enable comparisons with interventions for other diseases.”

2. Table 2. The additional details in the manuscript and Table 2 (PPV, true CIN2+ cases detected and colposcopy) are helpful. However, the information on colposcopy in Table 2 is a bit laborious for the reader to interpret as presented (some percents, some #s). Consider including both Ns and %s for the columns of women receiving colpo and women without CIN2 receiving colpo, or clarify interpretation of these columns with a table footnote to improve clarity.

We have added these to Table 2 and amended errors in the numbers for those receiving colpo without CIN2+, which added to the difficulty in interpreting these numbers.

3. Thank you for noting that a sensitivity analysis for VIA/VILI and cytology were conducted. The author response suggests this is included in Figure 2, but Figure 2 appears to be reporting the univariate analysis for careHPV. Please clarify where this information can be found.

Apologies that this update didn’t make it into the last submission. Figure 2 now includes VIA/VILI and cytology.

4. Time horizon: thanks for adding the time horizon. Suggest explicitly noting whether costs/outcomes were discounted (assuming not because the time horizon was less than one year).

This has been added to line 253: “Costs and outcomes were not discounted.”

5. Line 446-448: "While our analysis indicates that it is cost-effective compared to VIA alone, it is hard to ignore the capability of DNA testing to detect 97% of cases.” Not clear what "it is cost-effective compared to VIA alone" is referring to. If careHPV, should it be "less cost-effective compared to VIA alone" to align with the results? Please clarify.

We have clarified that we are referring to VIA/VILI: “While our analysis indicates that VIA/VILI is cost-effective compared to VIA alone and careHPV, it is hard to ignore the capability of DNA testing to detect 97% of cases.”

6. Line 459-460: "Loss to follow up circumvents the health gains that screening provides, consequently increasing the costs." Agree with the first part of this statement, however, LTFU may not necessarily result in increased costs when looking across the lifetime and evaluating cervical cancer cases averted (i.e., increased linkage to care often = higher costs associated with treatment). Suggest revising for clarity.

We have revised this for clarity. It now reads: “Loss to follow up circumvents the health gains that screening provides, and a lifetime model would be needed to capture the costs and health outcomes associated with it.”

---

## [Decision Letter · Decision Letter 2]

8 Mar 2021

Costs and cost-effectiveness of cervical cancer screening strategies in women living with HIV in Burkina Faso: the HPV in Africa Research Partnership (HARP) study

PONE-D-20-04078R2

Dear Dr. Devine,

We’re pleased to inform you that your manuscript has been judged scientifically suitable for publication and will be formally accepted for publication once it meets all outstanding technical requirements.

Kind regards,

Bruce A. Larson

Academic Editor

PLOS ONE

Additional Editor Comments (optional):

Reviewers' comments:

Reviewer's Responses to Questions

**Comments to the Author**

1. If the authors have adequately addressed your comments raised in a previous round of review and you feel that this manuscript is now acceptable for publication, you may indicate that here to bypass the “Comments to the Author” section, enter your conflict of interest statement in the “Confidential to Editor” section, and submit your "Accept" recommendation.

Reviewer #1: All comments have been addressed

Reviewer #2: All comments have been addressed

2. Is the manuscript technically sound, and do the data support the conclusions?

Reviewer #1: Yes

Reviewer #2: Yes

3. Has the statistical analysis been performed appropriately and rigorously? 

Reviewer #1: Yes

Reviewer #2: Yes

4. Have the authors made all data underlying the findings in their manuscript fully available?

Reviewer #1: Yes

Reviewer #2: Yes

5. Is the manuscript presented in an intelligible fashion and written in standard English?

Reviewer #1: Yes

Reviewer #2: Yes

6. Review Comments to the Author

Reviewer #1: (No Response)

Reviewer #2: The authors have adequately addressed all comments and limitations of the study, and the manuscript is now recommended for publication.

7. PLOS authors have the option to publish the peer review history of their article (what does this mean?). If published, this will include your full peer review and any attached files.

Reviewer #1: **Yes: **Julio C Teixeira

Reviewer #2: No

---

## [Editor Report · Acceptance letter]

10 Mar 2021

PONE-D-20-04078R2 

Costs and cost-effectiveness of cervical cancer screening strategies in women living with HIV in Burkina Faso: the HPV in Africa Research Partnership (HARP) study 

Dear Dr. Devine:

I'm pleased to inform you that your manuscript has been deemed suitable for publication in PLOS ONE. Congratulations! Your manuscript is now with our production department. 

Kind regards, 

on behalf of

Dr. Bruce A. Larson 

Academic Editor

PLOS ONE